# Current State of Bone Adhesives—Necessities and Hurdles

**DOI:** 10.3390/ma12233975

**Published:** 2019-11-30

**Authors:** Kai O. Böker, Katharina Richter, Katharina Jäckle, Shahed Taheri, Ingo Grunwald, Kai Borcherding, Janek von Byern, Andreas Hartwig, Britt Wildemann, Arndt F. Schilling, Wolfgang Lehmann

**Affiliations:** 1Department of Trauma Surgery, Orthopaedics and Plastic Surgery, University Medical Center Goettingen, Robert Koch Straße 40, 37075 Göttingen, Germany; katharina.jaeckle@med.uni-goettingen.de (K.J.); shahed.taheri@med.uni-goettingen.de (S.T.); arndt.schilling@med.uni-goettingen.de (A.F.S.); Wolfgang.Lehmann@med.uni-goettingen.de (W.L.); 2Fraunhofer Institute for Manufacturing Technology and Advanced Materials (IFAM), Wiener Straße 12, 28359 Bremen, Germany; katharina.richter@ifam.fraunhofer.de (K.R.); kai.borcherding@ifam.fraunhofer.de (K.B.); andreas.hartwig@ifam.fraunhofer.de (A.H.); 3Industrial and Environmental Biology, Hochschule Bremen—City University of Applied Sciences, Neustadtswall 30, 28199 Bremen, Germany; i.grunwald@hs-bremen.de; 4Ludwig Boltzmann Institute for Experimental and Clinical Traumatology, Austrian Cluster for Tissue Regeneration, Donaueschingenstrasse 13, 1200 Vienna, Austria; Janek.von.Byern@univie.ac.at; 5Faculty of Life Science, University of Vienna, Core Facility Cell Imaging and Ultrastructure Research, Althanstrasse 14, 1090 Vienna, Austria; 6Department 2 Biology/Chemistry, University of Bremen, Leobener Straße 3, 28359 Bremen, Germany; 7Experimental Trauma Surgery, University Hospital Jena, 07747 Jena, Germany; Britt.Wildemann@med.uni-jena.de

**Keywords:** bone, adhesive, glue, fracture, bioadhesive, osteosynthesis

## Abstract

The vision of gluing two bone fragments with biodegradable and biocompatible adhesives remains highly fascinating and attractive to orthopedic surgeons. Possibly shorter operation times, better stabilization, lower infection rates, and unnecessary removal make this approach very appealing. After 30 years of research in this field, the first adhesive systems are now appearing in scientific reports that may fulfill the comprehensive requirements of bioadhesives for bone. For a successful introduction into clinical application, special requirements of the musculoskeletal system, challenges in the production of a bone adhesive, as well as regulatory hurdles still need to be overcome. In this article, we will give an overview of existing synthetic polymers, biomimetic, and bio-based adhesive approaches, review the regulatory hurdles they face, and discuss perspectives of how bone adhesives could be efficiently introduced into clinical application, including legal regulations.

## 1. Bone Fractures and Osteosynthesis

Fractures are among the most common diagnoses that require inpatient hospitalization [1]. Bone fracture treatment consists of repositioning of the broken bone and fixation, so that bone healing can take place. In ancient history, this fixation was achieved conservatively from the *outside* through cast treatments. Since the beginning of the 1960s, the use of implants for *internal* fixation or osteosynthesis has largely superseded these conservative therapeutic approaches. The aim of therapeutic osteosynthesis after bone fracture is to restore the functional capacity of the bone as quickly as possible, in order to allow early mobilization of the patient [2,3,4], which is known to aid healing of the fracture [5].

Bone healing occurs in four phases: hematoma formation, early inflammatory, repair, and late remodeling phase [6]. After bone fracture, a hematoma develops within the fracture site. In the early inflammatory stage, inflammatory cells, mesenchymal stem cells (SCs), and fibroblasts infiltrate the bone. New vessels are formed while chondrocytes and fibroblasts lay down a collagen matrix that leads to callus formation around the fractured bone, generating early mechanical support [7]. Until this early stabilization, osteosynthetic support is necessary to shield the healing tissue from excessive external forces. Adequate strength comparable to properties before the injury is then typically achieved after 3–6 months [8]. Callus remodeling and finally, bone remodeling form the last stages of bone healing, which ideally lead to the restoration of the original shape and structure.

Surgical treatment of the fractures usually starts with a reduction of the fracture to bring the bone fragments in their original position and into close contact with each other, which maximizes the ability of the fracture surfaces to join and fuse. This is necessary for the bones to regain their ability to sustain strain [2]. This configuration then needs to be stabilized either from the outside with a cast and/or with implants such as screws, plates, and wires. 

## 2. Disadvantages of Current Internal Fixation

Metallic implants, as one major example for current internal fixation devices, must be removed in the vast majority of cases by additional surgeries. Thus, the patient is again faced with common risks that are inevitable in any surgical procedure. The removal of osteosynthesis materials is one of the most common surgical procedures in the western world [9]. The average costs amount to ∼800 € per out-patient care and ∼2000 € per short-term stationary interventions, weighing up to a total cost of ∼250 million € per year on the German healthcare system [9]. In 2010, about 180,000 metal removal procedures on the musculoskeletal system were performed in German hospitals alone, excluding implant removals by general practitioners [10]. At the University Medical Center in Göttingen, an annual average of 274 inpatients and approximately 624 outpatients with hand fractures were treated with osteosynthesis from 2012 to 2018. The average stay in the hospital was 7.7 days. During the same time period, 124 inpatients and 76 outpatients received a metal-based implant hand fracture treatment, including metal removal. The average stay of the operated patients was about 8.2 days. In addition to the expanded stay at the hospital, the cost of the metallic implants in conjunction with a wrist fracture adds up to 500–800 € for a standard supply with one plate, 5–6 angle stable screws distally, and three conventional screws in the shaft area. In the case of a standard restoration in the metacarpus area, the costs amount to approximately 171–300 € for using a plate with three screws each proximal and distal to the fracture. Summarized, the total costs for metallic implants at the University Medical Center in Göttingen were estimated at 137,000–219,200 € (22,833–36,533 €/year) for inpatients and 312,000–499,200 € (52,000–83,200 €/year) for outpatients with hand fractures for the mentioned time period, as an example for a typical large hospital. Standard restoration costs of the metacarpus area were approximately 46,854–82,200 € for inpatients and 106,704–187,200 € for outpatients.

Before the implant has to be removed, it has been reported that increased abrasion of metal implants may lead to unstable fracture supply and fracture-healing disorders in about 5% of the cases [11].

The infection rate of an osteosynthetic restoration is in the range of 1%–5% for closed fractures with elective surgeries, and it may increase up to 29%–40% in the case of higher-grade fractures [12]. These numbers are only based on the evaluations with currently available detection tools. Further improvement in diagnostic technology may even further increase this number [13].

Treatment of infected implants is time-consuming and tedious. On the surface of implants, bacteria are able to evolve from a planktonic form to a sessile phenotype. In the planktonic form, they divide very rapidly, causing an immune response that leads to clinical symptoms [14]. In this state, bacteria still have a high sensitivity to antibiotics. Once the phenotype evolves into the sessile form, both the metabolic and the division rate of the bacteria markedly slow down. In this state, a biofilm is formed that shields the bacteria from attacks by antibodies, activated phagocytes, and antibiotics [15,16]. The resulting biofilm is up to 1000 times less sensitive to the antibiotic therapy in comparison with the planktonic bacteria. In addition, more antibiotic resistant strains are generated after the phenotypic switch [13,17]. It appears that the body’s own immune system is only able to efficiently eliminate the planktonic forms [18,19], allowing sessile-induced infections to spread to adjacent tissues. This spreading results in abscess and/or fistula formation, osteitis, and potentially even sepsis. The patient’s immune response seems to be influenced by the implant material as well as the implant design [18,19]. The most common bacteria of implant-induced infections like *Staphylococci* contain receptors in their cell wall that enable them to adhere firmly to artificial material surfaces. The affinity of these bacteria to bone cement has been reported to be 15 times higher than to stainless steel and four times higher than to polyethylene [20]. Although one important property of bone adhesives is biodegradation, the aspect of bacterial infections should not be neglected. Since bone adhesives are injected and need to be stable for several weeks, the risk of infection should be a consideration in the development of new bone adhesives. 

Diagnosis and identification of the infecting bacterial strain depend on biological samples that are taken from the patients, coupled with sensitive molecular tools such as polymerase chain reactions and DNA sequencing [15,21]. However, clinical research on implant-related infections is hampered by the fact that both the planktonic and the sessile forms of many infectious bacteria are hard to culture *in vitro* [22]. 

## 3. Possible Clinical Advantages of Bone Adhesives

According to DIN EN 923:2016-03, an adhesive is a non-metallic substance capable of bonding materials by surface adhesion and its internal strength (cohesion). Conferring to the definition in DIN EN 923, bone cement is considered an adhesive, too. Nevertheless, we will focus in this review on materials different from classical bone cements, as the use of novel bone adhesives instead of currently used materials could solve several of the above-mentioned problems. Bone cement is a hardening material that is implanted together with the prostheses for anchoring functions. Thus, it serves mainly to stabilize the implant and is touched upon in this context mainly for comparison with newer concepts of bone adhesives.

From a mechanical perspective, the adhesive could act on the entire surface, providing a uniform areal distribution of the physical forces. It could also overcome certain disadvantages of metallic implants (e.g., physical stress and tissue damage in the area of bone fracture healing), which are associated with their relatively high stiffness and rigidity when compared to the bone material. Especially for low and medium-loaded body parts like carpus or metacarpus, bone adhesives may be a suitable alternative. Theoretically, broken fragments could be easily and directly joined together, and the adhesive could be gradually replaced by the re-growing bone [11]. This would prevent the necessity for secondary interventions, and also prevent the associated additional costs for the healthcare system. The costs of such a bone adhesive system are difficult to estimate so far since the production capacities and material costs are mainly unknown. Nevertheless, no metal implants would be needed, and operation times would presumably be shortened due to eliminating secondary interventions, bone preparation, and adaptation. Due to the reduced surgery time and the application of materials with higher possibilities to design biocompatibility, fewer complications such as infections, wound healing disorders, thrombosis, embolism, allergies, and intolerances can be expected. 

The required time for assessing the fracture and wound closure is similar in both osteosynthesis and bone adhesive surgery approaches and takes 30–60 min for non-complicated fractures, and at least 90 min for metal plate osteosynthesis in the area of the radius and the hand (Figure 1). 

Use of metal implants usually requires drilling the screw holes and adjusting the osteosynthesis plate, thus, increasing the procedure time. One operating hour costs about 302 € in Germany (2019), covering only the surgical team costs. In addition, operating and structural costs (surgical consumables, administration, operation room maintenance, etc.) are accounted with 16.63 € per minute, wrapping up to 998 € per hour [23].

In summary, treating bone fractures with bone adhesives would be an attractive alternative to metallic implants [24], especially for small fragment fractures like basal fractures of the thumb or Rolando fractures (Figure 1), debris fractures, and low-stressed bone regions. Bone adhesives have the potential to shorten the treatment time and partially eliminate subsequent treatments. Taken together, there is a high clinical demand for bone adhesive bonding technology. 

## 4. Necessary Properties for Bone Adhesives

The requirements for a medical grade bone adhesive are challenging. A clinically usable bone adhesive has to (1) be sterile and a biocompatible medical device, (2) stabilize the fracture sufficiently, (3) not impair bone healing and should be (4) easy to apply even at hard to reach areas, and (5) ideally be degraded after completion of bone healing without forming toxic metabolites.

(1) Sterility and biocompatibility for medical device classification

Medical devices are regulated by state authorities. In the United States, this is done by the Food and Drug Administration (FDA). Within the European Union (EU), the Medical Devices Regulation (MDR) 2017/745 defines the rules and requirements to bring a medical device to market. One of the most important and determining factors of the regulatory requirements for the manufacture is the device risk class. The type and duration of body contact determine the class, which ranges from I to III. The higher the device risk class is, the more biocompatibility and clinical testings are required. According to the MDR, all devices are classified whether they are non-invasive, invasive, or active devices. Based on this, combined with the application and interactions with the human body, the product class is to be determined (Annex IX of the MDR). Bioresorbable bone adhesives would be classified into the highest device class; III. This classification is derived from rule 8 of the MDR:

“… long-term surgically invasive devices are classified as class IIb unless […] they have a biological effect or are wholly or mainly absorbed, in which case they are classified as class III”.

Biological assessments are required whenever there is a direct physical contact, as in the case of bone adhesives. Test recommendations exist for this purpose. For example, biocompatibility is regulated in the ISO 10993 standard series “Biological evaluation of medical devices”; in particular, the biological effects and the resorption of a bone adhesive have to be reviewed with respect to the ISO 10993. A bone adhesive is an implant device, which is in contact with tissue and bone for a permanent duration (>30 days). For such a combination, the standard recommends considering aspects such as cytotoxicity, sensitization, irritation or intracutaneous reactivity, systemic toxicity (acute), subchronic toxicity (subacute toxicity), genotoxicity, and implantation.

Depending on the bone adhesive, the tests should be conducted at least by the following international standards:-DIN EN ISO 10993-3 Tests for genotoxicity, carcinogenicity and reproductive toxicity-DIN EN ISO 10993-5 Tests for *in vitro* cytotoxicity-DIN EN ISO 10993-6 Tests for local effects after implantation-DIN EN ISO 10993-9: Framework for identification and quantification of potential degradation products-DIN EN ISO 10993-10 Tests for irritation and skin sensitization-DIN EN ISO 10993-11 Tests for systemic toxicity.

In addition to DIN EN ISO 10993, the FDA has provided a Guidance for Industry and Food and Drug Administration Staff (FDA-2013-D-0743-0031, November 2016), in which the following biological effect categories are recommended to be reviewed too: Chronic Toxicity, Carcinogenicity, and Biodegradation.

Depending on the bone adhesive, the tests could be conducted by following international standards: -DIN EN ISO 10993-13: Identification and quantification of degradation products from polymeric medical devices-DIN EN ISO 10993-16: Toxicokinetic study design for degradation products and leachables-DIN EN ISO 10993-17: Establishment of allowable limits for leachable substances-DIN EN ISO 10993-18: Chemical characterization of medical device materials within a risk management process-And maybe other parts depending on the used system.

In addition to the tests, according to the ISO 10993, biocompatibility can also be assessed by critically reviewing the scientific literature. Here it must be ensured that the literature is up-to-date, relevant, and representative, and the equivalence to the device must be given. Only after the confirmation of the biocompatibility, the product can be tested for its effectiveness in the clinical trial, which is also prescribed by regulatory authorities. 

Within the complex field of regulatory requirements, it is important that the strategies for biocompatibility testing are reviewed separately for each medical device, considering the market, application, and current legislations to properly address the questions of device classification and biocompatibility testing rights.

For *in vivo* use, bone adhesives need to be sterile biomaterials. According to the American National Institute of Health, biomaterials are “substances or combinations of substances, other than therapeutically active substances, of synthetic or natural origin, that support or completely replace tissue, organ, or body functions for a certain period of time in order to maintain or improve an individual’s quality of life” [25]. Depending on the sterilization process, the uncured adhesive must withstand heavy gamma/beta radiations or the process of autoclaving and retain its adhesive properties, or a process would have to be developed in a clean room environment to guarantee sterility.

(2) Necessary mechanics for fracture stabilization

An adhesive is a non-metallic material that bonds parts together by surface adhesion (adhesion) and internal strength (cohesion). Adhesion is defined in DIN EN 923 “Adhesives—Terms and definitions” and in DIN 16920 “Adhesives, adhesive processing, terms”. It includes terms that are specific in context with adhesives and adhesive processing industries or which are generally used in adhesives application. Both adhesion and cohesion are necessary, so the adhesive can hold the bony parts of the fracture ends in place. In general, bone adhesives can be classified as structural adhesives. With regard to mechanical stresses, bone adhesives are subjected to different types of stress [26]. Depending on the location, the mechanical loads usually vary uniaxially between the respective proportions of static loads (shear, compression, tensile, shear, and torsional loads) and dynamic loads (vibration loads, as well as impact loads). As a general rule, the adhesive should be able to withstand peak loads, although the mechanical requirements for healthy bones are not necessarily taken as a guide value [27]. For osteosynthesis materials, a highest mechanical stress of approx. 100 MPa (1 MPa = 1 Million Pa = 1 N/mm^2^; 1 N is equivalent to 102 g) was observed on the anterior bridge around the outer hole, while the highest “on-screw” stress was located in the first screw at the anterior bridge with 150 MPa [28]. Similar values have been described as a key requirement for tissue adhesives for meniscus repair. The elastic modulus for this type of adhesive should be around 40–150 MPa, while the adhesive shear strength should be higher than 50 kPa [27].

There are numerous other standards in connection with sampling (DIN EN 1066), sample preparation, characterization (DIN EN 924, DIN EN 1067, DIN EN ISO 9665, DIN EN ISO 14678, DIN 53787, DIN 53788), and testing (DIN EN 1067, DIN EN ISO 9665, DIN 16945, DIN 53260, DIN 55405) of adhesives. Testing of tissues is regulated in ASTM F2255-05 (2010) and ASTM F2258-05 (Reapproved 2010).

In the past, attempts have been made to systematize adhesives according to certain criteria irrespective of their medical functionality [29]. This is common for industrial adhesives, and therefore, is briefly discussed here. These classifications can be based on mechanical properties (Figure 2) or the curing mechanisms, according to their composition or application. Adhesives are classified according to their curing mechanism in (i) chemical/reaction adhesives: polyaddition (epoxides, polyurethanes), polycondensation (silicones, silicon modified polymer adhesives, phenolic resins), polymerisation (cyanacrylates, methylmethacrylates, anaerobically and radiation curing adhesives), (ii) physical: via cooling (hotmelts), via evaporation of liquids (solvent- or water-based adhesives), and (iii) without solidification (pressure-sensitive adhesives).

Furthermore, immersion in liquid media brings in additional challenges for adhesive and cohesive properties. The discontinuous composition, the pH, and the omnipresence of moisture interfere with the initial adhesion and curing of many synthetic adhesive classes. At the same time, the physiological environment offers advantages for biocompatible materials, as well. The environment allows the materials to be metabolized in the body and be permanently integrated or degraded during wound healing. Nevertheless, the requirements for mechanical properties such as hardness, permeability, and elasticity in conjunction with the biological prerequisites (e.g., good osteoconductivity, non-inflammatory response, and no immune reaction) are extremely complex. 

Although bone cements are widely used for implant fixation in various orthopedic and trauma surgery applications, their suitability to be used as a bone adhesive is a matter of debate. Different approaches have been taken in order to overcome the lack of adhesive properties, local tensions at the adhesively bonded region, and the unsuitability of bone cements for delicate fractures (e.g., sternal bone fixation). This includes investigations on polyacrylate composites reinforced with inorganic fillers [31,32,33,34,35,36], a polyurethane system filled with hydroxyapatite [37], as well as biomimetic approaches, such as Kryptonite^TM^ (Doctors Research Group Inc, Southbury, CT, USA); a castor oil-derived polyurethane-based cement [38,39,40,41,42]. The latter is the first clinical adhesive to be used for joining bones in sternal closure [43,44]. Compared to PMAA, superior pullout strength in spine pedicle models, as well as superior peak bending and compressive yield stresses, were observed in cadaveric radius fractures. These data promised a valuable bone adhesive for fracture care. Nevertheless, Kryptonite^TM^ failed in the clinical routine and was recalled by Health Canada in 2012 for product safety concerns [45]. Preclinical testings were mainly performed on cadaveric models at ambient temperatures, and further studies revealed a reduction of strength and stiffness of approximately 50% when conducted at 37 °C. Furthermore, the US FDA recalled Kryptonite^TM^ in 2012 as well, due to the long and clinically unsuitable hardening time for the adhesive, which was initially 30 min followed by a further hardening of 24 h. 

Although different approaches in terms of biomimetic synthetic and hybrid concepts have been pursued to prepare bone adhesives [24,46,47,48], to this date, there has been no commercially available bone adhesive that meets all the above-mentioned requirements [49], but we are getting closer to wide clinical applications and are curious for mid- and long term results.

(3) Adhesives and bone healing

Another requirement for medical grade bone adhesives is not to hinder bone healing through remodeling. This is crucial regarding bonding bone fragments since the applied adhesive should ideally be absorbed and replaced by newly generated bone material. Due to the importance as well as the challenging nature of producing remodeling-friendly adhesives, this requirement will be elaborated in more detail.

At first glance, the skeleton appears to be static. However, it is a dynamic and living structure that adjusts to its necessities in a variable environment [50]. During development and up until adult age, bones continue to grow at growth zones called epiphyseal plates, located at the endings of the long bones [51,52]. Epiphyseal plates contain cartilage cells (i.e., chondrocytes), which enable the bone growth. Afterward, minerals are deposited and cause ossification of the chondrocytes, thereby generating firm bone [52]. Bones constantly undergo reconstructions, referred to as “bone remodeling”, in order to adjust to changing circumstances such as loss or gain of body weight, bone fractures, and increase or decrease of muscle mass [51,52]. Bone remodeling is not only an ongoing process but also serves the repair of bone defects such as fractures. The process depends on the physiological environment of the bone, for example, in response to increasing strains on the bone, the structure is strengthened, and when the strain decreases, it gets less stable [53]. 

Bone remodeling involves two primary cell types, osteoblasts and osteoclasts [54,55]. Osteoblasts are responsible for the production of bone tissue, more precisely, the organic portion of the bone matrix (osteoid). The second cell type, the osteoclasts, are antagonists of the osteoblasts. They prevent bone overgrowth through bone resorption. This antagonistic system acts in coordination, i.e., osteoblasts generate bone primarily in places where the osteoclasts have removed old bone tissues [55]. The interaction is based on signaling between the two cell types. Osteoblasts produce a diffusible ligand molecule that acts on the RANK receptor located at the cell surface of the osteoclasts. RANK ligand binding causes the activation, differentiation, and survival of the osteoclasts [56,57]. A balanced equilibrium is provided by a second molecule (again) produced by the osteoblasts. This factor, osteoprotegerin, is a receptor molecule inserted into the cell membrane of the osteoblasts, which can also bind the RANK ligand, thereby decreasing the RANK ligand concentration. This results in the reduction of the RANK ligand-RANK receptor interaction, subsequently decreasing the osteoclast activity [58]. 

A third cell type (osteocytes) participates in bone remodeling by operating the process in response to external pressures. These cells receive and provide information by sensing mechanical stresses and transfer the information via biochemical signals, such as sonic hedgehog [59,60]. Osteocytes are interconnected to each other via long cytoplasmic extensions that occupy tiny canals (canaliculi) serving as the exchange pathway of nutrients and cellular wastes [61].

Bone remodeling plays a key role in fracture healing after complete separation of two or more bone segments. Two different types of healing processes can be distinguished: (i)The primary or direct healing depends on the direct contact of the fractured parts and proceeds without callus formation. This process involves the formation of bone multicellular units (BMUs) by osteoclasts and subsequent apposition of newly generated bone substances by the osteoblasts, in which the two ends of the fracture are extended towards each other and fuse. After about eight weeks, trajectories are formed in response to compressive and tensile loadings, shaping the bone accordingly [62].(ii)The secondary or indirect type of healing refers to fractures that leave a gap between different parts of the fractured bone. This tissue will be subsequently invaded by chondroblasts. The resulting fibrocartilage undergoes ossification by the osteoblasts, forming a callus area which is mechanically less stable than the intact bone tissue [62]. Finally, the callus is converted to bone tissue mimicking the lines of the trajectories. Depending on the severity of the fracture and individual characteristics of the patients, the entire fracture healing process can take between six months to one year [62,63].

(4) Application

The application of bone adhesives is a crucial and important aspect. Since time and space are limited during the operation procedure, we propose a fast application by a cannula system on the bone fracture surface. This allows the minimal invasive use of the bone adhesive. The application time should not exceed 10 min, because operation time is valuable, and anesthesia time should be minimized. Aside from one-component adhesives, two-component systems could also be used, since they have been proven to be successful in the past (e.g., Tisseel fibrin glue) [64,65]. Pre-treatment of bone surfaces by acidic substances could be an advantage but should be restricted to several seconds incubation time depending on the substance, since surrounded tissues can be damaged by prolonged contact times [66,67]. Further treatments like UV curing appear suitable if short exposures and long wavelengths (>315 nm, UV-A) are used. Shorter wavelengths can lead to tissue damage [68].

(5) Biodegradation

A possible advantage of bone adhesive could be biodegradation. Ideally, the bone adhesive degrades in a way that it is replaced by bone tissue. This might be by chemical degradation under physiological conditions or biochemically (e.g., increased hydrolysis by enzymes). The disintegration of the adhesive by osteoclasts, whereby callus can be formed and later converted into bone tissue, would be another suitable replacement mechanism [69,70]. The adhesive degradation must be phased with bone formation, while the degradation process and the resulting degradation products should definitely not interfere with bone remodeling. The time-dependent coordination between adhesive degradation and bone formation is hard to achieve [71,72], making further studies to find suitable adhesives that fulfill all these characteristics necessary. In any case, swelling of the adhesive by the aqueous environment is a prerequisite for any degradation reaction under physiological conditions.

## 5. Development of Approaches for the Generation of Bone Adhesives

Already in the 1940s, Hedri presented the first bone adhesives based on gelatin, epoxy resins, or acrylates [73]. For different reasons, these adhesives are not appropriate. This includes low adhesion on wet surfaces, low biocompatibility, and formation of non-biodegradable residues. Since then, intensive research has been carried out to improve this approach. Detailed information regarding the chemistry of different types of bone-adhesive materials and ensuing mechanisms of adhesion can be found elsewhere [49]. The main lines of research can be classified into synthetic, biomimetic, and biobased approaches:

(1) Synthetic adhesives

Fully synthetic formulations (e.g., polyacrylic acid [74,75,76] or polyester [77]) account for the most frequently studied type of bone adhesives due to their tunability of adhesive features, cross-linking degree, functional groups, and their viscosity [49]. 

So far, the adhesive class of epoxies has not been convincing due to biocompatibility issues, despite having promising mechanical strengths [24]. Polyurethanes, which have also long been considered biocompatible, have performed very well in recent studies. For example, a polyurethane foam reinforced with hydroxyapatite crystals was successfully developed to bind bone to bone. In comparison to bone cement, this showed an adhesion (tensile strength) that was four times higher on unprimed bone, while the adhesion to primed bone was only two times higher [37]. 

Methacrylates and cyanoacrylates belong to the adhesive class, which are considered to have a high potential for bonding to bone. Based on the data from experiments with pig bones carried out with commercially available “super glues”, it was shown that they did not hold as well as screws. Nevertheless, the following values were obtained, all within a similar size range: 1.22 ± 0.50 MPa Histoacryl^®^, 1.16 ± 0.43 MPa Super Bonder^®^ and 1.70 ± 0.45 MPa (NeoOrtho^®^ Screws) [78]. Due to their properties, cyanoacrylates are very promising for bonding bones [79].

On account of these characteristics, cyanoacrylates have been further developed to achieve micro tensile and shear bond strengths in the range of 1 to 2 MPa; the latter for N-butyl cyanoacrylate is significantly greater than a plate and screw reference [80]. Wistlich et al. reported a photochemically curing system consisting of a poly(ethylene glycol) dimethacrylate matrix and a six-armed isocyanate functional starshaped prepolymer that, in situ, undergoes an interpenetrating network enriched with biodegradable ceramic fillers. The use of 20–40 wt% isocyanate (NCO) terminated prepolymer improved the adhesion strength to cortical bone after storage in buffered saline by a factor of 2 (0.15–0.2 to 0.3–0.5 MPa) [81]. The histological and biomechanical evaluation of N-butyl-2-cyanoacrylate as a bone adhesive in an animal study showed that the adhesive is as effective as plates and screws in surgically fabricated osteotomy fixations [82]. Likewise, histomorphometric data from the fixation of a cortical bone graft confirmed a lower bone formation in the case of the adhesive compared to screws [83].

Another example from the acrylate toolbox refers to nanobioactive glass fillers incorporated into a polymer, based on polypropylene fumarate and hydroxyethyl methacrylate (HEMA). This redox-curable composite was used for bonding ribbon bones and showed a tensile and shear strength of about 9 MPa [84].

Additionally, a Thiol-Ene reaction was applied for bone adhesive. The adhesive network is formed by a thiol-induced crosslinking that attacks unsaturated double bonds, forming an elastic polymer. The thiol groups were provided as tris[2 -(3-mercapto-propionyloxy)ethyl]isocyanurate, and double bonds were provided in 1,3,5-triallyl-1,3,5-triazine-2,4,6(1H,3H,5H)-trione. A multi-stage application of a primer, a layer-by-layer process (adhesive–fiber–adhesive), and light curing (λ = 400 nm) resulted in a shear bond strength of 9.0 MPa. Based on preclinical trials with rat femur fractures and porcine metacarpal fractures, this adhesive system withstood loads of up to 70 N for 1000 cycles, was biocompatible, and did not induce inflammatory reactions [85,86]. Nevertheless, the application must be simplified in order to obtain a suitable surgical product.

The most promising synthetic adhesives organized by adhesion strength are summarized in Table 1. The use of adhesives for bonding fractures could revolutionize surgical procedures for personalized bone repair. However, there has been no commercially available adhesive to use as a pure bone adhesive so far. Likewise, synthetic adhesives have not been able to reach the mechanical requirements of 40–150 MPa mentioned earlier. Furthermore, insufficient biocompatibility, adhesive strength, and fixation protocols are the biggest obstacles that need to be overcome [85]. To address these issues, biomimetic approaches have been proposed.

(2) Biomimetic approaches

Around 100 marine and terrestrial organisms are known to secrete adhesives [87]. Such living beings have developed their bioadhesives over 500 million years of evolution to meet specific requirements. The functions vary and range from settlement, hunting, and defense to locomotion. Nevertheless, the mechanical properties, production, composition, and secretion of the vast majority of bioadhesives have not yet been clarified.

The most famous representatives of marine and terrestrial organisms that secrete bioadhesives are the marine sandcastle worm *Phragmatopoma californica* and the mussel *Mytilus edulis*, the sea cucumber, or terrestrial species as the Australian frog *Notaden bennetti*. The sandcastle worm uses its adhesive to build tubes, which serve as its dwelling. It has four secretory cell types for this purpose, which contributes to form a coacervate. The coacervate forms the cement between the grains of sand and the tube (which consists of sand itself). This is a classic hybrid material of sand and secretion, which can be managed very economically with small amounts of the secreted adhesive. This type of adhesive is suitable for underwater applications [88], making this hybrid natural model an obvious material choice for bone adhesives. This can range from coacervates made from a mixture of aminated gelatin and polyphosphodopamide [89] to Tetranite^TM^ developed by Professor Ken Gall and co-workers at Duke University, USA [90]. Tetranite^TM^ is based on O-phospho-L-serine; a component of many proteins that exist in natural secretions. It is bioresorbable with an instant adhesive strength, fulfills almost all requirements for a bone adhesive, and is currently undergoing approval by the FDA. Based on O-phosphoserine and tetracalcium phosphate, a bone adhesive has been developed, which is curable in an aqueous environment in a matter of minutes, and provides high bone-to-bone adhesive strength [48]. The presented material has a higher shear strength up to ten times compared to calcium phosphate cements and PMMA bone cements. Both systems have already been tested and are used in many clinics. In addition, chemical adhesion to titanium surfaces is reported [48]. This should be twice as high as the binding to bone material. Moreover, a 52-week study with a distal demur size defect was performed in rabbits. Tetranite^TM^ degradation started at week eight and continued during the whole experiment, ending up with 77.5% degradation after 52 weeks in the femoral condyle of rabbit. Furthermore, signs of osteointegration, bone ingrowth, and resorbability were observed via histology slices. These results propose Tetranite^TM^ as a bioresorbable adhesive that promotes osteointegration due to its degradation features [48].

In connection with the phosphoserine-based approach, Pajari-Palmer and colleagues reported a “Novel Class of Injectable Bioceramics” composed similarly to TetraniteTM. Instead of Tetracalciumphosphate, however, α-Tricalciumphosphate was used in combination with phosphoserine, which were curable under humid conditions and showed up to 40 times the bond strength (2.5–4 MPa) of commercial cyanoacrylates (0.1 MPa), and were 100-fold stronger than surgical fibrin adhesives (0.04 MPa) [91]. The use of cyanoacrylate bone adhesive in mandibular angle fracture fixation was studied [92]. It was concluded that they were useful systems to reduce complications in non-load-bearing areas.

The mechanical properties of mussel adhesives include strong adhesion to almost any kind of surface up to 10 MPa [93,94], low modulus of elasticity of 0.9 GPa vs. 1.2 GPa for collagen from mammalian tendon, and approx. 3 GPa for human femur [95] and residual resilience of 53% after fatigue test (collagen from mammalian tendon 90% [94]). This would also have qualified mussel adhesive as bone adhesives; however, their functionality is based on an extremely complex interaction of different proteins [96,97]. So far, it has not been possible to achieve the unique strengths of mussel adhesives by abstracting the protein motifs. Nevertheless, a commercial mussel adhesive has been developed [98]. This was obtained by pre-modified intestinal bacteria using an enzyme derived from *Methanocaldococcus jannaschii*. During biosynthesis, this enzyme inserted a protective group for the reactive 3,4-dihydroxyphenylalanine (DOPA); one of the key amino acids in the mussel proteins. For this purpose, the bacterium was fed with the already protected amino acid ortho-nitrobenzyl DOPA. The deprotection was then carried out photochemically by irradiation with light [98]. In addition to the DOPA’s deprotection by irradiation, a photochemical curing mechanism (photo crosslinking technique of tyrosine residues by oxidation, λ = 452 nm) for recombinant produced mussel proteins with DOPA has been described earlier [99]. Apart from photochemical cross-linkage, mussel-inspired adhesives can be crosslinked reliably using oxidation agents, such as iron. Iron-induced networks of a mussel-inspired adhesive showed sufficient bond for sternal closure while having biodegradability, low cytotoxicity, and low exothermicity [43]. Furthermore, the suitability of mussel-inspired adhesives for joining titanium implants was simulated, in terms of osteogenic differentiation of osteoblasts on titanium mesh coated with musselpeptide-RGD, and positive results were revealed in terms of osteostimulation. The *in vitro* blood response increased compared to bare titanium mesh. Bone regeneration in a rat calvarial defect was manifested by a significant increase of bone volume (42% coated vs. 25% bare titanium, respectively) and bone area (160 mm^2^ coated vs. 118 mm^2^ bare titanium, respectively). These observations were supported by the histological analysis in terms of the corresponding bone thicknesses (0.75 mm coated vs. 0.47 mm bare titanium), and relative bone mineral density (95% of normal bone) [100]. Based on this, it was demonstrated that priming the bone with a layer-by-layer technique coated with a mussel-based material improved adhesion and cell response [101,102]. Other bioinspired approaches include the work of Malkoch and his team on allyl, methacrylamide, and thiol groups. The developed adhesive showed shear strength values of about 0.3 MPa [101].

The mentioned biomimetic adhesives are prearranged according to their adhesion strength in Table 2. These organisms have been studied for decades and have already enriched the market with products that can be traced back to them. In the meantime, exploring other organisms beyond the sandcastle worm, and the mussel has opened up new perspectives; the results provide information on further bioadhesives such as *Notaden* frog adhesive [103], the goose barnacles *Dosima fascicularis* [104], or even from North American salamander as *Plethodon shermani* [105].

(3) Biobased adhesives

Although a sharp distinction between biomimetic and biobased adhesives cannot necessarily be made, many sugar- and protein-based bone adhesives are subsumed under natural adhesives.

A proteinogenic, autologous fibrin adhesive was successfully tested as an alternative to K-wire in rabbits [106]. Beyond proteinogenic adhesives, sugar-based components are also candidates for bone adhesives. This was confirmed by the studies on two-component chitosan and dextran bone adhesive hydrogels [107,108]. A similar but covalently cross-linked system was also based on chitosan and contained calcium carbonate. It formed hydrogels and revealed increased underwater strength over time in the presence of calcium carbonate while simultaneously showing excellent adhesions at all times tested. Finally, it was observed that the hydrogel lead to normal cell growth and good cell differentiation on the adhesive surface, making it a promising candidate for the treatment of bone fractures [46].

Incorporation of calcium carbonate and hydroxyapatite in biocomposites to be used as chitosan-based bone adhesives is another example of biobased approaches. It was shown that coordination bonds between calcium ions and chitosan amino groups were formed, thus producing ionic cross-links between chitosan macromolecules that altered chitosan gelation mechanisms, gel strength, and adhesion in bone surfaces [109]. The calcium carbonate concentration seems to be a key factor for these phenomena. A formulation containing 2% chitosan and 4% calcium carbonate and hydroxyapatite showed the most promising performance, with high adhesion to the bone surface (0.27 MPa), and a cohesive failure mode (failure in the adhesive material, not at the interface between surface and adhesive). Formulations with higher levels of calcium carbonate and hydroxyapatite showed the most promising values for the intended application [109]. The revealed biomimetic adhesives are represented in Table 3.

## 6. Adhesives for Tendon-to-Bone Repair

Adhesives are not only developed for the connection of bone-to-bone but also to connect bone and soft tissue. The connection between soft and hard material, however, is always a challenge, and in the musculoskeletal system, this connection is called enthesis, which connects tendon/ligament (T/L) to bone. Two types of enthesis exist: the fibrous and the fibrocartilage enthesis. The transition zone of the fibrocartilage enthesis consists of bone, unmineralized and mineralized fibrocartilage and T/L [110]. The mechanical stresses on this structure can be very high and result in a deformation of the T/L, but almost no deformation of the bone. Also, the tensile moduli of these tissues are very different: 200 MPa for tendon and a 100-fold higher for bone. Collagen type 1, however, is the main extracellular matrix protein in both structures [111]. Due to these deformation/stress differences, the transition zone between hard and soft tissue is prone to failure. For example, rotator cuff tears are the result of a failing enthesis. This complex structure has only a limited regeneration potential and heals with a scar, resulting in a high re-tear rate [110]. In recent years, improved suture techniques have been developed with better initial mechanical stability, but also with a risk of tissue trauma caused by the suturing material leading in failure and different clinical outcomes [112]. Therefore, healing of the transition zone might be supported by a better fixation of the tendon to the bony footprint. Since the various suture techniques have not resulted in a significant clinical improvement, other approaches are needed, and gluing of the tendon to the bone is a promising approach. The following summarizes various approaches:

The potential of a synthetic adhesive has been investigated in a human cadaveric model for the repair of rotator cuff tears. Loctite 4903 only, an ethyl and octyl cyanoacrylate adhesive, resulted in a weak maximum strength, whereas the suture with and without the adhesive exhibited much higher maximum strengths [113]. However, no *in vitro* cytocompatibility and *in vivo* healing data are available.

The use of fibrin as a clotting substance dates back to 1909, while in 1972, a fibrin sealing for nerve anastomosis was described, and in the 1980s, fibrin glue was also used to support bone healing [114]. An early study from 1989 showed good histological results after augmenting tendon-to-bone healing with fibrin sealant without an accelerated inflammatory reaction in a dog model [115]. The suture of the rabbit Achilles tendon insertion was augmented with fibrin glue with or without growth factors. A biomechanically improved healing was seen by the pure fibrin glue with a further enhancement due to the applied BMP-2 [116]. In a cohort study, significantly fewer re-tears were observed after augmentation of rotator cuff repair with adipose mesenchymal stem cell loaded fibrin [116].

A magnesium-based bone adhesive (MBA) has been used in experimental animal models to augment T/L-to-bone repair with conflicting results. The repair of the anterior cruciate ligament (ACL) was improved by the MBA in a rabbit model as assessed by histology and improved load to failure after six weeks [117]. While MBA in a dog study also improved initial mechanical properties of the flexor tendon-to-bone stability, *in vivo* healing for 21 days was impaired due to an enhanced inflammatory or allergic reaction to the adhesive used [118].

Millar et al. investigated the possibility of supporting tendon-to-bone stability in a cadaveric rotator cuff repair model. They used an adhesive from the Australian frog and showed significantly improved initial strength of the repaired rotator cuff, regardless of the used suture technique [119].

## 7. Outlook and Conclusions

During the last few decades, numerous efforts have been set into the development and adaptation of a bone adhesive. So far, no product that combines all important properties is on the market. The main reason for the failure in adaptation and production of a bone adhesive, so far, seems to lie in the specific and challenging requirements of the bone environment. Current bone adhesives cannot combine the three important requirements: biocompatibility, degradability, and bond strength. Nevertheless, the first promising approaches are now in clinical testing, and results can be expected in the near future. Current advanced technologies like genetic engineering, tissue engineering, or biotechnology approaches open further possibilities for bone adhesive research. Although it seems unlikely that conventional osteosynthesis will be replaced by bone adhesive anytime soon, the research on new adhesive technologies is an important prospective for the future.

## Figures and Tables

**Figure 1 materials-12-03975-f001:**
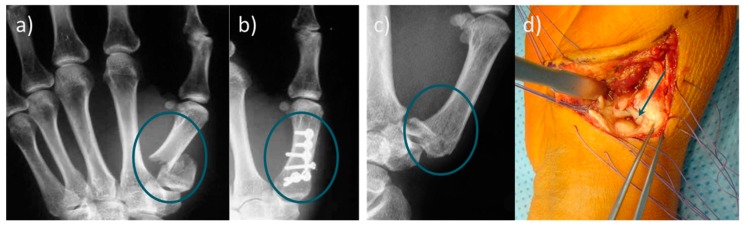
(**a**) Radiograph of a basal fracture of the thumb. (**b**) Conventional osteosynthesis of the basilar fracture; (**c**) radiological image of a Rolando fracture, and (**d**) representation of the small bone fragments in the joint area, which can only be fixed unsatisfactorily by classical osteosynthesis. Refixation of metacarpal fractures is usually complicated and time-consuming. Since limitations of conventional osteosynthesis exist, the use of bone adhesives would be particularly useful in the metacarpus area.

**Figure 2 materials-12-03975-f002:**
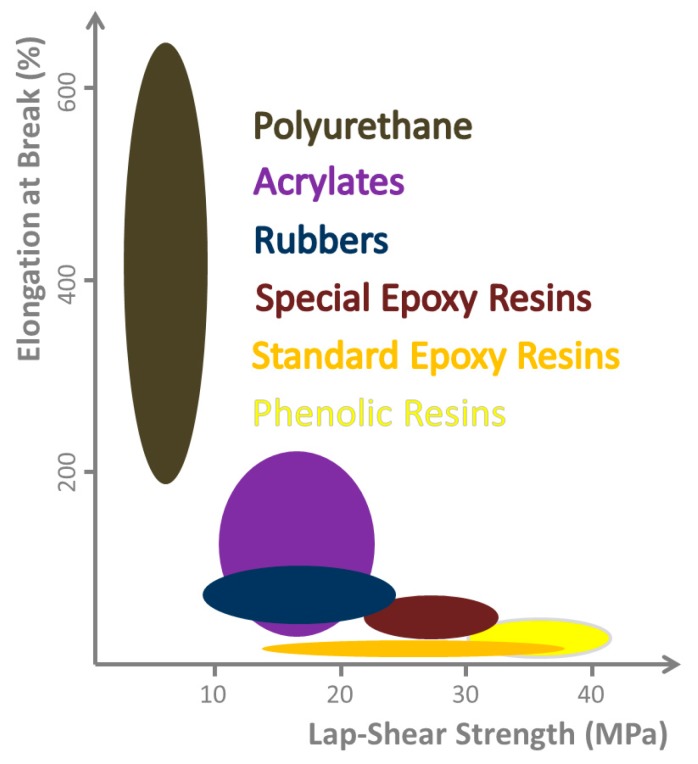
Range of mechanical properties obtainable with different chemical classes of adhesives. Relation between lap-shear strength and elongation at break, as they cannot be optimized independently. A case related compromise must be found, but in general, both properties should be as high as possible. (Illustration based on [30]).

**Table 1 materials-12-03975-t001:** Summary of synthetic adhesives arranged by adhesion strength.

Synthetic Adhesive	Adhesion Strength	Published
Thiol-Ene reaction based polymers	9.0 MPa	Granskog et al., 2018 [85] Arseneault et al., 2018 [86]
Nanobioactive glass fillers + HEMA	9.0 MPa	Shahbazi et al., 2016 [84]
Methacrylates and cyanoacrylates NeoOrtho^®^	1.70 ± 0.45 MPa	Vieira et al. 2016 [78]
Methacrylates and cyanoacrylates Histoacryl^®^	1.22 ± 0.50 MPa	Vieira et al. 2016 [78]
Methacrylates and cyanoacrylates Super Bonder^®^	1.16 ± 0.43 MPa	Vieira et al. 2016 [78]
Poly(ethyleneglycol) dimethacrylate matrix + isocyanate	0.3–0.5 MPa	Wistlich et al., 2017 [81]

**Table 2 materials-12-03975-t002:** Comparison of biomimetic adhesives organized by adhesion strength.

Biomimetic Adhesives	Adhesion Strength	Published
Tetranite^TM^	62 ± 8 MPa	Kirillova et al., 2018 [48]
Mussel adhesives	10 MPa	Price, 1981 [93]; Gosline, 2002 [94]
Injectable Bioceramics	2.5–4 MPa	Pujari-Palmer et al. 2018 [91]
Activated dopamine derivatives	0.3 MPa	Olofsson et al., 2016 [101]

**Table 3 materials-12-03975-t003:** Evaluation of biobased adhesives sorted by adhesion strength.

Biobased Adhesive	Adhesion Strength	Published
Chitosan and dextran bone adhesive	0.39 MPa	Balakrishnan et al., 2017 [108]
Chitosan, calcium carbonate and hydroxyapatite combinations	0.27 MPa	Pinzón et al., 2017 [109]
Proteinogenic, autologous fibrin adhesive	n.a.	Azarpira et al., 2017 [106]

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
