# Peer review of "Current State of Bone Adhesives—Necessities and Hurdles"

_materials, 2019, doi:10.3390/ma12233975_

Round 1

Reviewer 1 Report

Thank You for submitting this interesting manuscript, which briefly and concisely summarizes the biological background of the development of bone-glues as substitutes for osteosynthesis screws and plates.

The manuscript also highlights comprehensively the long way from theory to experimental setups ex and in vivo and mandatory clinical studies until a final reliable product can be handed out to the clinician and the different procedural approaches and complexity of developing a reliable "bone-glue.

The authors mentioned Kryptonite in line 271 as an example of a commercially available product, but forgot to mention that this product was withdrawn again in 2012 due to failure in "real life" since bonding-studies were performed only on human cadavers and application procedure was too long for clinical use.  (Ball, C. G., Grondin, S. C., Pasieka, J. L., Kirkpatrick, A. W., MacLean, A. R., Cantle, P., ... & Hamilton, M. (2018). Examples of dramatic failures and their effectiveness in modern surgical disciplines: can we learn from our mistakes?. Journal of comparative effectiveness research7(7), 709-720.)

A short explanation of why Kryptonite failed in the clinical routine application might even add a better comprehension of why the development of bone-glues is a very complex matter.

Summary: this reviewer recommends the submitted manuscript for publication in the revered journal "Materials"

Author Response

Dear Reviewer 1,

Thank you very much for giving us the opportunity to revise our manuscript “Current state of bone adhesives - necessities and hurdles”

Manuscript ID: materials-642215

We have now adapted our text according to your suggestions. The respective changes are labelled inside the manuscript. We want to thank you for your effort and feel that this allowed us to further improve the review.

We hope that the text now meets the high standards of “Materials” and look forward for your reply.

Please find the detailed answers in the attachment.

Kind regards

Kai Böker

Reviewer 2 Report

In ‘Current state of bone adhesives – necessities and hurdles’ Böker et al. review bone adhesives (synthetic, biomimetic, and bio-based) as well as legal and regulatory issues towards their clinical application. They situate the clinical problem as well as possible clinical advantages of bone adhesives and the disadvantages of current internal fixation methods. They then detail 5 key properties that their believe are necessary for bone adhesives. Finally, they provide an overview of the synthetic, biomimetic, and bio-based materials that have been developed as bone adhesives as well as bone-to-tendon adhesives. Overall, the review recapitulates many of the concepts in a recent review (cited in this manuscript - Sánchez‐Fernández, María J.; Hammoudeh, Hussein; Félix Lanao, Rosa P.; van Erk, Machteld; van Hest, Jan C. M.; Leeuwenburgh, Sander C. G.: Bone‐Adhesive Materials. Clinical Requirements, Mechanisms of Action, and Future Perspective. In: Adv. Mater. Interfaces 6 (4), 2019, S. 1802021. DOI: 10.1002/admi.201802021) so further clarification of the novelty of the present review is warranted.

As a general comment, it would be helpful to have a clear definition of what is considered a bone adhesive for the purpose of this review in the beginning of the manuscript.

As another general comment, some further thought could be given to the organization of the section on ‘Development of approaches for the generation of bone adhesives’. Is this meant to give an overview of all of the current approaches, those that have shown the best experimental results, or those that are the most recent? Overall, this section seems to give a number of examples, but they seem rather randomly organized so it is unclear if this is a thorough coverage of the literature on the topic.

Specific comments:

Lines 106-108: Is bone cement considered to be a bone adhesive? Further, what is the relevance of the discussion of biofilm formation if the ultimate goal is to develop materials that are degraded?

Lines 245-250: Are these the adhesives of interest for the review? It seems that most of this list is composed of synthetic polymers that are in many cases not degradable? How are these biomimetic or bio-based?

Lines 333-344: This section on the application of the bone adhesives should first be written in a paragraph form, and additional information (and supporting references) should be provided about the different requirements that are listed.

Lines 345-352: Likewise, the section on biodegradation should be expanded. Biomaterials degrade by different mechanisms so having a requirement for degradation only be osteoclasts seems rather limited.

Lines 363-373: This part of the text is unclear. What is meant by the vulcanization concept? Is this adhesive network formed by sulfur-induced crosslinking meant to be an example of this? Is it at thiol-ene reation?

Line 433: What is meant by stickiness value?

Lines 441-444: Can more details about this formulation be provided?

Line 450: This line is unclear. What do the percentages mean?

Lines 466-467: More information is needed. How was the simulation performed? Was it supported with experimental data?

Lines 473-476: Should this be deleted as it is included in the section on tendon-bone adhesives?

Line 475: Do you mean approved for clinical use?

Lines 497-506: How is this example of a chitosan, calcium carbonate, and hydroxyapatite combination different from the system described in lines 490-496, which seems to have a similar composition?

Author Response

Dear Reviewer 2,

Thank you very much for giving us the opportunity to revise our manuscript “Current state of bone adhesives - necessities and hurdles”

Manuscript ID: materials-642215

We have now adapted our text according to your suggestions. The respective changes are labelled inside the manuscript. We want to thank you for your effort and feel that this allowed us to further improve the review.

We hope that the text now meets the high standards of “Materials” and look forward for your reply.

Please find the detailed answers in the attachment.

Kind regards

Kai Böker

Round 2

Reviewer 2 Report

The authors have addressed my previous concerns with their revision.